

# Obesity in the Balinese is associated with FTO rs9939609 and rs1421085 single nucleotide polymorphisms

Lidwina Priliani[1], Sukma Oktavianthi[1], Ria Hasnita[2,3],
Hazrina T. Nussa[4], Rut C. Inggriani[5], Clarissa A. Febinia[1],
Anom Bowolaksono[4], Rini Puspitaningrum[2], Rully A. Nugroho[5],
Ketut Suastika[6] and Safarina G. Malik[1]

[1] Eijkman Institute for Molecular Biology, Jakarta, Indonesia
[2] Faculty of Mathematics and Natural Sciences, Universitas Negeri Jakarta, Jakarta, Indonesia
[3] Department of Surgery, Faculty of Medicine, Universitas Indonesia, Jakarta, Indonesia
[4] Department of Biology, Faculty of Mathematics and Natural Sciences, Universitas Indonesia, Depok, Indonesia
[5] Faculty of Biology, Satya Wacana Christian University, Salatiga, Indonesia
[6] Division of Endocrinology and Metabolism, Department of Internal Medicine, Faculty of Medicine, Udayana University, Denpasar, Indonesia

## ABSTRACT

Obesity prevalence is increasing worldwide, including in the Bali Province, Indonesia, a popular tourism destination area. The common single nucleotide polymorphisms (SNPs) rs9939609 and rs1421085 of the fat mass and obesity-associated (FTO) gene have been repeatedly reported as one of the important obesity genetic risk factors. We have examined the associations of FTO rs9939609 and rs1421085 SNPs with obesity in the 612 unrelated Balinese subjects living in urban and rural areas. Linear and logistic regression analyses with adjustment for age and gender were employed to investigate the association between FTO genotypes, haplotypes and obesity parameters. We found that the FTO SNPs genotypes increased BMI by 1.25 kg/m$^2$ ($p = 0.012$) for rs9939609 AA and 1.12 kg/m$^2$ ($p = 0.022$) for rs1421085 CC, particularly in females and in rural population. Subjects carrying these genotypes also showed a tendency to maintain high BMI, regardless of their age. Our result showed that the FTO rs9939609 and rs1421085 risk alleles were associated with increased BMI and obesity in the Balinese.

## INTRODUCTION

Rapid transitions in lifestyle and diet towards excessive consumption of energy-dense food and reduction of physical activity result in rising obesity prevalence worldwide (*World Health Organization, 2018*). According to the National Basic Health Survey, obesity prevalence in Indonesia increases by 4.3% from 2007 to 2013, and 7% from 2013 to 2018, respectively (*RISKESDAS, 2007, 2013, 2018*). As a known leading risk factor for chronic non-communicable diseases (such as hypertension, type two diabetes mellitus, cardiovascular diseases, fatty liver, stroke, and some types of cancers), obesity contributes

Corresponding author
Safarina G. Malik, ina@eijkman.go.id

to economy and health burdens (*Must et al., 1999*; *Basen-Engquist & Chang, 2011*; *Al-Goblan, Al-Alfi & Khan, 2014*). As the risk of these chronic diseases increases, quality of life decreases and health care costs escalate (*Withrow & Alter, 2011*; *Cawley & Meyerhoefer, 2012*).

The complex interplay between environmental and multiple genetic factors that influence body mass index (BMI) has been proposed as a trigger of increased obesity and its comorbidities, with heritability estimated to be around 40–70% (*Herrera & Lindgren, 2010*). Among significant obesity genetic risk factors, the common single nucleotide polymorphisms (SNPs) rs9939609 and rs1421085 in the fat mass and obesity-associated (FTO) gene have been consistently reported associated with obesity in distinct populations (*Chang et al., 2008*; *Cha et al., 2008*; *Fawwad et al., 2015*; *Babenko et al., 2019*).

The Balinese population has undergone rapid lifestyle transition from traditional to modern lifestyle in line with escalating economic growth in the region, most likely due to rapid development in tourism industry (*Antara & Sumarniasih, 2017*). Several studies showed the adoption of Western diet, reduction of physical activities, increasing socioeconomic status and education are associated with increased adiposity (*Huntsman, White & Gunung, 2005*) and obesity prevalence in urban population (*Suastika et al., 2011b*). Obesity prevalence in Bali was at 15.5%, higher than the national obesity prevalence at 14.8% in 2013 (*RISKESDAS, 2013*). The interplay between genetics and lifestyle factors has been found to associate with obesity and its comorbidities in Balinese population (*Malik et al., 2011*, *Oktavianthi et al., 2012*). In this study, we examined the association of FTO rs9939609 and rs1421085 with obesity in Balinese of the Bali Province, Indonesia. We hypothesize that FTO gene variants play a role in elevating BMI and obesity risk in the Balinese.

# MATERIALS AND METHODS

## Subjects, study design, measurements

A cross-sectional study enrolling 612 participants from five locations (286 female and 326 male) in the Bali Province, Indonesia, was conducted in 2008–2015 with written informed consent (*Malik et al., 2011*; *Suastika et al., 2011b*; *Oktavianthi et al., 2012*, *2018*). The five locations represents urban (Legian and Denpasar) and rural (Penglipuran, Nusa Ceningan and Pedawa) settings, based on Statistics Indonesia's criteria (*Badan Pusat Statistik, 2010*). The map and sample size of the five locations are shown in Fig. S1. Ethical approvals for this study were granted by the Eijkman Institute Research Ethics Commission (No. 32 on 27 October 2008 and No. 80 on 24 December 2014), and by the Faculty of Medicine Ethic Committee, Universitas Udayana (No. 690a/SKRT/X/2010 on 28 October 2010 and No. 1286/UN.14.2/Litbang/2014 on 18 September 2014). Collected demographic and anthropometric data include: age, gender, weight, height, and waist circumference (WC). BMI was calculated as weight in kg divided by (height)$^2$ in m$^2$, while waist to height ratio (WHtR) was calculated as WC divided by height, both measured in the same unit. WHtR is a proxy for central (visceral) adipose tissue (*Swainson et al., 2017*). The high BMI (BMI ≥ 25 kg/m$^2$) and high WC (male ≥ 90 cm;

female ≥ 80 cm) cut offs were according to the Asia–Pacific perspective redefining obesity in adult Asian, while the high WHtR cut off at ≥ 0.5 was based on previous reports (*Lee et al., 1995*; *Hsieh & Yoshinaga, 1995*; *WHO Regional Office for the Western Pacific, 2000*).

### DNA extraction and genotyping

Genomic DNA was extracted as previously described (*Malik et al., 2011*). The FTO rs9939609 and rs1421085 variants were detected using amplification-refractory mutation system (ARMS) polymerase chain reaction (PCR). Detection of rs9939609 was performed using a previously published primer sets (*Fawwad et al., 2015*) while detection of rs1421085 was done using a novel primer sets, designed using Primer1 (*Collins & Ke, 2012*) and BioEdit® Sequence Alignment Editor (Ibis Bioscience, Carlsbed, CA, USA). Lists of outer and inner primers for ARMS-PCR is described in Table S1. Optimation of the annealing temperature was done using the Veriti® 96 West Thermal Cycler (Applied Biosystem, Foster City, CA, USA), while the ARMS-PCR was performed using GeneAmp® PCR System 9700 (Applied Biosystems, Foster City, CA, USA). ARMS-PCR conditions are described in Table S1. PCR products were resolved on 2% agarose gel electrophoresis (Lonza, Basel, Switzerland). Confirmation of variant alleles were carried out by DNA sequencing using BigDye® Terminator v.3.1 Cycle Sequencing Kits, with ABI 3130xl Genetic Analyzer (Applied Biosystem, Foster City, CA, USA).

### Statistical analysis

Statistical analysis were carried out in R version 3.4.0 (www.r-project.org) with R Studio version 1.0.143 (www.rstudio.com). The five sampling sites (four villages and one city) map was generated from Google Static Maps using the "ggmap" and "ggrepel" packages. Continuous variables are presented as mean (±SD). All SNPs were tested for departure from Hardy–Weinberg equilibrium (HWE). Genotype distributions and linkage disequilibrium (LD) between SNP pairs were calculated using the "genetics" package (*Warnes et al., 2019*). Haplotypes were determined using expectation maximization (EM) algorithm as implemented in the "haplo.glm" function of the R "haplo.stats" library. Genetic associations analyses were conducted using both linear and logistic regression models with adjustments for age and gender (male/female). Significant level based on Bonferroni correction was set at 0.025 (*p* value = 0.050/2) (*Nichols & Hayasaka, 2003*).

## RESULTS

### The Balinese characteristics

Characteristics of the study subjects are summarized in Table 1. The male to female ratio was comparable (53.3% vs. 46.7%). The Balinese showed an elevated mean of obesity parameters (BMI 24.0 ± 4.83 kg/m$^2$; WC 83.9 ± 11.6 cm; WHtR 0.53 ± 0.07). Genotype distribution for rs9939609 is TT 34%, TA 49% and AA 17%; while for rs1421085 is TT 37%, TC 46%, and CC 18%. Minor allele frequency (MAF) for both variants were 0.42 and 0.41, respectively (Table 1). The rs9939609 and rs1421085 presented a high linkage disequilibrium with D' = 0.90 and $r^2$ = 0.88.
**Table 1 Characteristics and genotypes of studied subjects.** Data are presented as mean (±SD) for age, weight, height BMI, WC, WHtR, and $n$ (%) for gender and genotypes frequency. BMI, body mass index; WC, waist circumference; WHtR, waist to height ratio; MAF, minor allele frequencies; HWE, Hardy Weinberg equilibrium; LD, linkage disequilibrium.

| Variables | Total ($n = 612$) | Male ($n = 326$) | Female ($n = 286$) | Urban ($n = 318$) | Rural ($n = 294$) |
|---|---|---|---|---|---|
| Age (years) | 46.6 ± 14.6 | 47.9 ± 13.1 | 45.0 ± 16.0 | 42.8 ± 12.7 | 50.7 ±15.5 |
| Weight (kg) | 61.4 ± 14.6 | 66.3 ± 13.7 | 55.7 ± 13.6 | 66.7 ±14.5 | 55.6 ± 12.4 |
| Height (cm) | 159.0 ± 8.9 | 165.0 ± 7.7 | 154 ± 6.1 | 162.0 ± 8.4 | 157.0 ± 8.7 |
| BMI | 24.0 ± 4.8 | 24.4 ± 4.5 | 23.5 ± 5.1 | 25.4 ± 4.7 | 22.5 ± 4.5 |
| WC (cm) | 83.9 ± 11.6 | 86.5 ± 11.4 | 81.0 ± 11.1 | 87.5 ±11.2 | 80.0 ± 10.7 |
| WHtR | 0.5 ± 0.1 | 0.5 ± 0.1 | 0.5 ± 0.1 | 0.5 ± 0.1 | 0.5 ± 0.1 |
| FTO rs9939609* | | | | | |
| TT | 208 (34) | 103 (32) | 105 (37) | 116 (36) | 92 (31) |
| TA | 300 (49) | 160 (49) | 140 (49) | 138 (43) | 162 (55) |
| AA | 104 (17) | 63 (19) | 41 (14) | 64 (20) | 40 (14) |
| $p$-value HWE | 0.868 | 1 | 0.709 | 0.065 | 0.022 |
| MAF | 0.42 | 0.44 | 0.39 | 0.42 | 0.41 |
| FTO rs1421085* | | | | | |
| TT | 224 (37) | 114 (35) | 110 (38) | 126 (40) | 98 (33) |
| TC | 280 (46) | 146 (45) | 134 (47) | 135 (42) | 145 (49) |
| CC | 108 (18) | 66 (20) | 42 (15) | 57 (18) | 51 (17) |
| $p$-value HWE | 0.209 | 0.140 | 0.901 | 0.059 | 0.905 |
| MAF | 0.41 | 0.43 | 0.38 | 0.39 | 0.42 |
| LD Corr | 0.88 | 0.87 | 0.90 | 0.88 | 0.89 |
| D' | 0.90 | 0.89 | 0.92 | 0.93 | 0.90 |

**Note:**
* $n$ (%). The significant $p$-values are <0.05.

## The FTO rs9939609 and rs1421085 are associated with obesity

Of all the genetic models developed (Tables S2 and S3), the recessive model was the most suitable for the population in this study. In multiple linear regression analyses with recessive genetic model, the minor rs9939609 AA and rs1421085 CC genotypes increased BMI by 1.25 kg/m$^2$ ($p = 0.012$) and by 1.12 kg/m$^2$ ($p = 0.022$), respectively. Multiple logistic regression analyses further confirmed the trend of increased odds for higher BMI in subjects carrying the minor genotypes of both SNPs (odds ratio = 1.59, $p = 0.042$ for rs9939609; and odds ratio = 1.57, $p = 0.047$ for rs1421085). Of the non-genetic parameters, age was shown to influence BMI, and being male increased BMI and WC, while living in an urban setting increased all obesity parameters (Table 2).

To investigate the involvement of gender and environment in relationship to FTO SNPs and obesity parameters, we conducted separate analyses for male and female subjects (Tables S4 and S5), as well as for urban and rural populations (Tables S6 and S7). Significant associations between rs9939609 AA and the rs1421085 CC genotypes with increased BMI were only found in females in both linear (estimate = 1.97 kg/m$^2$ and $p = 0.021$ for rs9939609 AA, estimate = 2.35 kg/m$^2$ and $p = 0.005$ for rs1421085 CC

Table 2 **Associations of FTO SNPs with obesity**[*]. Statistical analysis was done using linear regression model, with adjustments for age, gender (male/female) and population (urban/rural). The model used for linear regression: outcome ~ SNPs + age + gender + population. BMI, body mass index; WC, waist circumference; WHtR, waist to height ratio. High BMI, BMI ≥ 25; High WC, male's WC ≥ 90 cm and female's WC ≥ 80 cm; High WHtR, WHtR ≥ 0.5. The significant $p$-values after Bonferroni correction are indicated in bold ($p < 0.025$).

| $n = 612$ | BMI | | WC | | WHtR | |
|---|---|---|---|---|---|---|
| | **Estimates** | $p$ | **Estimates** | $p$ | **Estimates** | $p$ |
| FTO rs9939609 | | | | | | |
| AA | 1.25 | **0.012** | 1.51 | 0.190 | 0.01 | 0.363 |
| Age (years) | −0.01 | 0.482 | 0.01 | 0.633 | <0.01 | **0.011** |
| Male | 0.82 | 0.029 | 5.26 | **<0.001** | <0.01 | 0.467 |
| Urban | 2.73 | **<0.001** | 7.49 | **<0.001** | 0.03 | **<0.001** |
| FTO rs1421085 | | | | | | |
| CC | 1.12 | **0.022** | 1.86 | 0.102 | 0.01 | 0.156 |
| Age (years) | −0.01 | 0.438 | 0.01 | 0.679 | <0.01 | **0.013** |
| Male | 0.82 | 0.029 | 5.24 | **<0.001** | <0.01 | 0.444 |
| Urban | 2.80 | **<0.001** | 7.56 | **<0.001** | 0.03 | **<0.001** |
| $n = 612$ | **High BMI** | | **High WC** | | **High WHtR** | |
| | **Odds ratio** | $p$ | **Odds ratio** | $p$ | **Odds ratio** | $p$ |
| FTO rs9939609 | | | | | | |
| AA | 1.59 | 0.042 | 1.31 | 0.228 | 1.17 | 0.506 |
| Age (years) | 1.00 | 0.800 | 1.01 | 0.260 | 1.00 | 0.546 |
| Male | 1.67 | **0.004** | 0.53 | **<0.001** | 0.94 | 0.714 |
| Urban | 3.14 | **<0.001** | 2.77 | **<0.001** | 2.93 | **<0.001** |
| FTO rs1421085 | | | | | | |
| CC | 1.57 | 0.047 | 1.52 | 0.060 | 1.49 | 0.090 |
| Age (years) | 1.00 | 0.745 | 1.01 | 0.294 | 1.00 | 0.600 |
| Male | 1.67 | **0.004** | 0.53 | **<0.001** | 0.93 | 0.664 |
| Urban | 3.22 | **<0.001** | 2.81 | **<0.001** | 2.96 | **<0.001** |

**Note:**
[*] Recessive model.

genotypes) and logistic (odds ratio = 2.83 and $p = 0.003$ for rs9939609 AA, odds ratio = 3.36 and $p = 0.001$ for rs1421085 CC genotypes) regression analyses (Table 3).

These genotypes also presented distinct effects on obesity parameters in different environment, as shown in Table 4. In urban, increased WC was associated with the rs9939609 AA genotype in linear regression analysis (estimate = 3.04 cm, $p = 0.038$), and with the rs1421085 CC genotypes in logistic (odds ratio = 1.95, $p = 0.037$) regression analyses. Meanwhile in rural, the rs9939609 AA showed a tendency to increase BMI by 1.53 kg/m$^2$, while the rs1421085 CC genotypes demonstrated a significantly increased BMI by 1.65 kg/m$^2$ ($p = 0.016$), and higher odds for high BMI (odds ratio = 2.25 and $p = 0.016$).

In haplotype analyses that incorporated a recessive genetic model, the AC haplotype consisting the minor alleles A of rs9939609 and C of rs1421085 was associated with obesity

**Table 3 Associations of FTO SNPs with obesity in male vs. female\*.** Statistical analysis was done using logistic regression model while adjusting for age and population (urban/rural). The model used for linear regression: outcome ~ SNPs + age + population. BMI: body mass index, WC: waist circumference, WHtR: waist to height ratio. High BMI: BMI ≥ 25; High WC: male's WC ≥ 90 cm and female's WC ≥ 80 cm; High WHtR: WHtR ≥ 0.5. The significant *p*-values after Bonferroni correction are indicated in bold (*p* < 0.025).

| | Male (*n* = 326) | | | | | | Female (*n* = 286) | | | | | |
|---|---|---|---|---|---|---|---|---|---|---|---|---|
| | BMI | | WC | | WHtR | | BMI | | WC | | WHtR | |
| | Estimates | *p* | Estimates | *p* | Estimates | *p* | Estimates | *p* | Estimates | *p* | Estimates | *p* |
| FTO rs9939609 | | | | | | | | | | | | |
| AA | 0.69 | 0.242 | 0.83 | 0.569 | <0.01 | 0.848 | 1.97 | **0.021** | 2.11 | 0.251 | 0.01 | 0.314 |
| Age (years) | −0.02 | 0.186 | −0.04 | 0.341 | <0.01 | 0.331 | −0.01 | 0.715 | 0.04 | 0.369 | <0.01 | 0.053 |
| Urban | 3.44 | **<0.001** | 9.32 | **<0.001** | 0.04 | **<0.001** | 1.91 | **0.003** | 5.53 | **<0.001** | 0.02 | **0.014** |
| FTO rs1421085 | | | | | | | | | | | | |
| CC | 0.23 | 0.688 | 0.61 | 0.670 | <0.01 | 0.850 | 2.35 | **0.005** | 3.48 | 0.056 | 0.02 | 0.067 |
| Age (years) | −0.02 | 0.168 | −0.04 | 0.323 | <0.01 | 0.337 | −0.01 | 0.693 | 0.04 | 0.395 | <0.01 | 0.059 |
| Urban | 3.49 | **<0.001** | 9.36 | **<0.001** | 0.04 | **<0.001** | 2.03 | **0.002** | 5.66 | **<0.001** | 0.02 | **0.010** |
| | High BMI | | High WC | | High WHtR | | High BMI | | High WC | | SS | |
| | Odds ratio | *p* | Odds ratio | *p* | Odds ratio | *p* | Odds ratio | *p* | Odds ratio | *p* | Odds ratio | *p* |
| FTO rs9939609 | | | | | | | | | | | | |
| AA | 1.02 | 0.948 | 1.11 | 0.73 | 1.08 | 0.795 | 2.83 | **0.003** | 1.61 | 0.185 | 1.22 | 0.586 |
| Age (years) | 0.99 | 0.359 | 1.00 | 0.991 | 0.99 | 0.388 | 1.00 | 0.871 | 1.01 | 0.26 | 1.01 | 0.204 |
| Urban | 4.43 | **<0.001** | 3.41 | **<0.001** | 3.62 | **<0.001** | 2.03 | **0.016** | 2.25 | **0.002** | 2.46 | **0.001** |
| FTO rs1421085 | | | | | | | | | | | | |
| CC | 0.90 | 0.716 | 1.32 | 0.343 | 1.53 | 0.175 | 3.36 | **0.001** | 1.85 | 0.085 | 1.41 | 0.344 |
| Age (years) | 0.99 | 0.361 | 1.00 | 0.958 | 0.99 | 0.359 | 1.00 | 0.850 | 1.01 | 0.270 | 1.01 | 0.214 |
| Urban | 4.45 | **<0.001** | 3.42 | **<0.001** | 3.63 | **<0.001** | 2.22 | **0.008** | 2.33 | **0.002** | 2.49 | **0.001** |

Note:
\* Recessive model.

parameters (BMI by 0.73 kg/m$^2$ (*p* = 0.008), WC by 1.47 cm (*p* = 0.022), respectively (Table S8). The AC haplotype demonstrated a tendency for increased WC by 1.99 cm (*p* = 0.041) in females (Table S9), and was also associated with increased BMI in rural population (Table S10).

## The FTO rs9939609 and rs1421085 maintained high BMI in subjects older than the mean age of ≥46.6 years

The mean age of this population is 46.6 years old (Table 1). Our result showed that the obesity parameters (BMI, WC and WHtR) in subjects <46.6 years have a tendency to increase, while in subjects ≥46.6 years, they have a tendency to decrease (Fig. 1). Interestingly, subjects carrying the homozygous variants of both SNPs (rs9930609 AA and rs1421085 CC genotypes) sustained their high BMI regardless of their age, in contrast to the wild-type and heterozygous genotypes carriers which demonstrated a trend towards decreasing BMI with age. Meanwhile, the relationships between WC and WHtR and age are not modulated by the FTO genotypes (Fig. 1).

Table 4 **Associations of FTO SNPs with obesity in urban vs. rural**[*]. Statistical analysis was done using logistic regression model while adjusting for age and gender (male/female). The model used for linear regression: outcome ~ SNPs + age + gender. BMI: body mass index, WC, waist circumference; WHtR, waist to height ratio. High BMI, BMI ≥ 25; High WC, male's WC ≥ 90 cm and female's WC ≥ 80 cm; High WHtR, WHtR ≥ 0.5. The significant $p$-values after Bonferroni correction are indicated in bold ($p < 0.025$).

| | Urban (n = 318) | | | | | | Rural (n = 294) | | | | | |
|---|---|---|---|---|---|---|---|---|---|---|---|---|
| | BMI | | WC | | WHtR | | BMI | | WC | | WHtR | |
| | Estimates | p | Estimates | p | Estimates | p | Estimates | p | Estimates | p | Estimates | p |
| FTO rs9939609 | | | | | | | | | | | | |
| AA | 1.09 | 0.097 | 3.04 | 0.038 | 0.02 | 0.088 | 1.53 | 0.043 | −0.59 | 0.743 | <0.01 | 0.694 |
| Age (years) | 0.03 | 0.192 | 0.14 | **0.003** | <0.01 | **<0.001** | −0.04 | **0.012** | −0.09 | **0.026** | <0.01 | 0.518 |
| Male | 1.28 | **0.021** | 6.22 | **<0.001** | <0.01 | 0.978 | −0.04 | 0.932 | 2.94 | **0.018** | −0.02 | 0.044 |
| FTO rs1421085 | | | | | | | | | | | | |
| CC | 0.63 | 0.361 | 2.03 | 0.189 | 0.01 | 0.326 | 1.65 | **0.016** | 1.68 | 0.304 | 0.01 | 0.286 |
| Age (years) | 0.03 | 0.213 | 0.14 | **0.004** | <0.01 | **<0.001** | −0.04 | **0.010** | −0.09 | **0.020** | <0.01 | 0.460 |
| Male | 1.32 | **0.018** | 6.29 | **<0.001** | <0.01 | 0.974 | −0.05 | 0.915 | 2.87 | **0.021** | −0.02 | **0.037** |
| | High BMI | | High WC | | High WHtR | | High BMI | | High WC | | High WHtR | |
| | Odds ratio | p | Odds ratio | p | Odds ratio | p | Odds ratio | p | Odds ratio | p | Odds ratio | p |
| FTO rs9939609 | | | | | | | | | | | | |
| AA | 1.36 | 0.285 | 1.71 | 0.075 | 1.69 | 0.141 | 2.10 | 0.044 | 0.92 | 0.828 | 0.84 | 0.625 |
| Age (years) | 1.01 | 0.585 | 1.03 | **0.008** | 1.04 | **0.001** | 0.99 | 0.160 | 0.99 | 0.291 | 0.98 | 0.042 |
| Male | 2.17 | **0.001** | 0.55 | **0.017** | 0.91 | 0.726 | 1.02 | 0.936 | 0.41 | **<0.001** | 0.72 | 0.168 |
| FTO rs1421085 | | | | | | | | | | | | |
| CC | 1.19 | 0.567 | 1.95 | 0.037 | 1.75 | 0.143 | 2.25 | **0.016** | 1.20 | 0.585 | 1.38 | 0.304 |
| Age (years) | 1.00 | 0.610 | 1.03 | **0.010** | 1.04 | **0.001** | 0.99 | 0.140 | 0.99 | 0.269 | 0.98 | 0.034 |
| Male | 2.19 | **0.001** | 0.54 | **0.015** | 0.91 | 0.733 | 1.02 | 0.951 | 0.41 | **<0.001** | 0.71 | 0.148 |

Note:
[*] Recessive model.

# DISCUSSION

Association studies of the FTO gene with obesity or obesity-related traits have been reported in many populations across the world, confirming the strong association of FTO SNPs with BMI and/or obesity (*Frayling et al., 2007*; *Scuteri et al., 2007*; *Hotta et al., 2008*; *Chang et al., 2008*; *Srivastava et al., 2016*). A meta-analyses study demonstrated the correlation between FTO rs9939609 and rs1421085 with obesity in Hispanic, Caucasian, and Asian populations (*Peng et al., 2011*).

In Indonesian population, most report on FTO rs9939609 SNP association with obesity came from the western part of the country, namely North Sumatera (*Lubis et al., 2017*), Yogyakarta (*Iskandar et al., 2018*), West Sumatera (*Susmiati, Surono & Jamsari, 2018*), and DKI Jakarta (*Daya et al., 2019*). Despite the diversity of the Indonesian population (*Karafet et al., 2010*; *Tumonggor et al., 2013*) and the differences in sociocultural exposure, our report from Bali, which is located in the central part of Indonesia, showed that the FTO variants are also genetic risk factors for obesity in the Balinese, similar to previously reported populations from the western part of Indonesia.

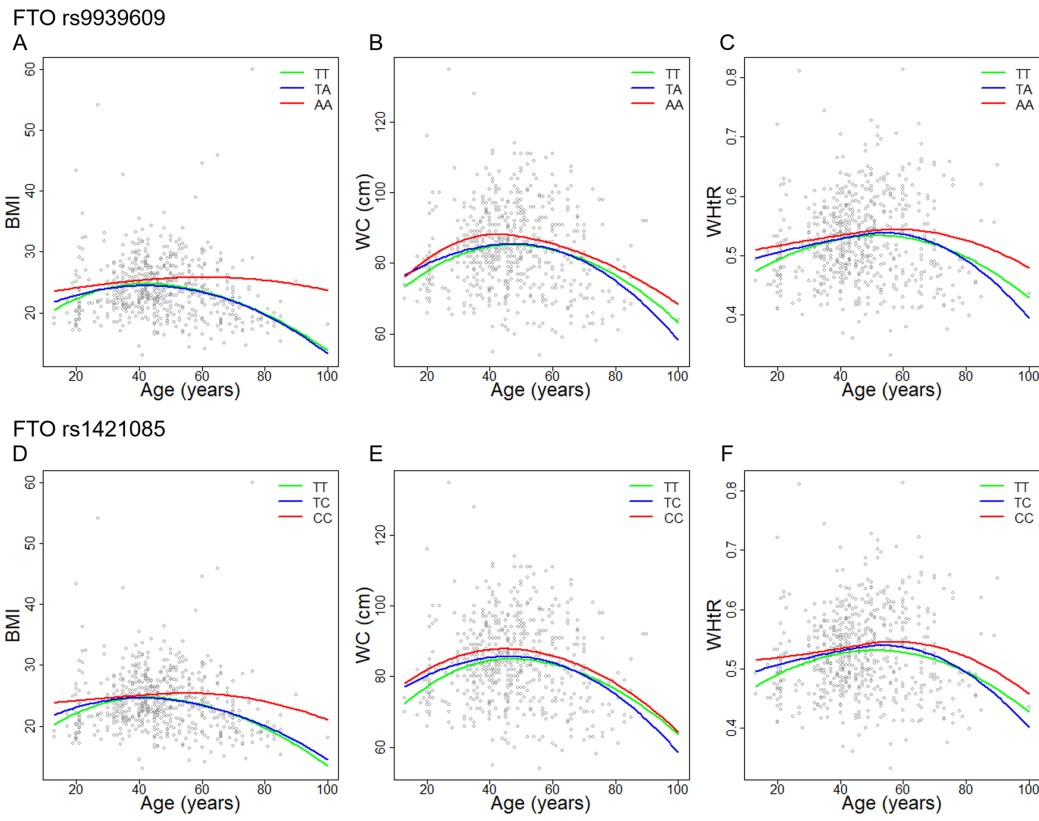

**Figure 1 Association between obesity parameters and age within FTO SNPs alleles.** FTO rs9939609 showing (A) BMI, (B) WC and (C) WHtR. FTO rs1421085 showing (D) BMI, (E) WC, and (F) WHtR.

In this study, we showed that having the FTO rs9939609 and rs1421085 SNPs risk alleles increase the risk for obesity. We have reported recently that individuals carrying the risk allele of rs9939609 demonstrated a higher risk to develop obesity due to preferences for high dietary fat intake (*Daya et al., 2019*). FTO might play a role in controlling feeding behavior, modifying energy expenditure (*Fawcett & Barroso, 2010*), reducing satiety responsiveness (*Wardle et al., 2008*), increasing consumption of highly palatable food (*Wardle et al., 2009*), and losing control over eating (*Tanofsky-Kraff et al., 2009*). Another study suggested a link between FTO, protein intake, and body weight (*Merritt, Jamnik & El-Sohemy, 2018*). All of these might influence the different effects of FTO on female vs. male, and urban vs. rural. In this study, we did not assess food intake, which is the limitation of this study.

Our current report indicated a gender-specific effect, where the associations of the A allele of rs9939609 and the C allele of rs14210845 with BMI were stronger in female than in male. These findings were consistent with previous studies in children and adolescents in Swedish and Chinese population (*Jacobsson et al., 2008*; *Zhang et al., 2014*). A meta-analysis study has found that 25 FTO SNPs including the rs9939609 and the rs1421085, were specifically associated with obesity in females only (*Tan et al., 2014*). This gender differences might be explained by the variation in body composition between males

and females (*Zillikens et al., 2008*). A comparative study of twins across eight countries reported that there is a gender difference in the heritability of BMI (*Schousboe et al., 2003*). However, other studies have reported that the rs9939609 was associated with BMI in both gender (*Frayling et al., 2007*; *Qi et al., 2008*). These discrepancies may be due to distinct genetic background, environmental factors, and sample sizes. Moderate sample size is another limitation of this study. Population-based studies with a larger sample size will be beneficial for further investigation of the possible interactions between FTO SNPs and genders.

In general Balinese population, BMI tend to decrease by age, however, our result showed that high BMI is maintained in individual carrying the FTO rs9939609 and rs1421085 risk alleles. This may imply the long-lasting effect of FTO risk alleles in increasing the obesity risk, regardless of the age. The basal metabolic rate decreases along with age and will lead to metabolic abnormality (*Henry, 2000*). Our previous study showed that metabolic decline is more prominent in older Balinese (*Suastika et al., 2011a*). Thus, awareness in maintain healthy lifestyle should begin from younger age to prevent obesity and its related comorbidities. A survey of nutritional habits in teenagers reported that eating errors (i.e., irregular eating, skipping breakfast) were more frequently observed in overweight and obese students as compared to the normal weight ones, and emphasized the importance of conveying the knowledge on the causes of overweight and obesity and the rules of healthy dieting (*Zalewska & Maciorkowska, 2017*).

Our previous studies in Balinese showed that the associations between genetic risk factors and obesity were different in urban dan rural area. In this study and our previous study of ADRB3 SNP showed that the association between genetic risk factor and obesity were found in rural (*Malik et al., 2011*). However, the association between UCP2 genetic risk factors and obesity was only found in urban (*Oktavianthi et al., 2012*). Discrepancies between urban and rural might be due to the influence of lifestyle and environmental exposures, as well as gene-environment interaction. Urban and rural have their own environmental characteristics. The characteristics of urban area include population density of ≥5,000 persons/$km^2$, less than 25% work in the agricultural sector and have more than eight public facilities (i.e., high school, health center, roads that can accommodate four wheeled motorized vehicles, factories, etc.). The characteristic of rural area include population density of <1,000 and most of the land are used for farming (*Mulyana, 2014*). All of these might influence their lifestyle, from being active and hardworking as farmers or fishermen into less active as hotel worker or small store owners. People living in urban area also consume more ready-to-eat food and their food variation are lower than people in rural area (*Ghaisani, 2017*).

Obesity is influenced by a complex interplay between multiple genes and environmental risk factors, such as consumption of high energy dense food and sedentary lifestyle. This powerful combination might predispose the high prevalence of obesity in urbans, which increased the risk to develop non-communicable diseases. Nevertheless, a recent report revealed that contrary to the major views, more than 80% of the global rise in mean BMI from 1985 to 2017 in low- and middle-income countries was the result of BMI increases in rural areas (*NCD Risk Factor Collaboration, 2019*).

## CONCLUSION

The Balinese population showed a high MAF of the FTO rs9939609 and rs1421085 risk alleles that were associated with increased BMI and obesity. Considering that these risk alleles could have a long-lasting effect in this population, knowledge on healthy lifestyle and diet should be introduced and endorsed not only to the urban Balinese, but also the rural population, although their average BMI are still within the normal range.

## ACKNOWLEDGEMENTS

The authors are grateful to all volunteers for their participation in this study. The authors thank Drs. Made Ratna Saraswati, Pande Dwipayana, Desak Made Wihandani, and I. Wayan Weta for their support during sample collections. We thank the field medical doctors, medical faculty students, clinical pathology laboratory and research assistants for their support in this study. We thank Dr. Ni Luh Made Agustini Leonita for her help in DNA isolation. We are grateful to Profs. Herawati Sudoyo and Sangkot Marzuki for their support and encouragements.

### Funding

The research was supported by the block grant from the Government of Republic of Indonesia through the Ministry of Research and Technology for the Eijkman Institute for Molecular Biology. The funders had no role in study design, data collection and analysis, decision to publish, or preparation of the manuscript.

### Grant Disclosures

The following grant information was disclosed by the authors:
Government of Republic of Indonesia through the Ministry of Research and Technology for the Eijkman Institute for Molecular Biology.

### Competing Interests

The authors declare that they have no competing interests.

### Author Contributions

- Lidwina Priliani analyzed the data, conceived and designed the experiments, prepared figures and/or tables, authored or reviewed drafts of the paper, and approved the final draft.
- Sukma Oktavianthi analyzed the data, conceived and designed the experiments, prepared figures and/or tables, authored or reviewed drafts of the paper, and approved the final draft.
- Ria Hasnita analyzed the data, performed the experiments, authored or reviewed drafts of the paper, and approved the final draft.
- Hazrina T. Nussa analyzed the data, performed the experiments, authored or reviewed drafts of the paper, and approved the final draft.

- Rut C. Inggriani analyzed the data, performed the experiments, authored or reviewed drafts of the paper, and approved the final draft.
- Clarissa A. Febinia analyzed the data, authored or reviewed drafts of the paper, and approved the final draft.
- Anom Bowolaksono analyzed the data, authored or reviewed drafts of the paper, and approved the final draft.
- Rini Puspitaningrum analyzed the data, authored or reviewed drafts of the paper, and approved the final draft.
- Rully A. Nugroho analyzed the data, authored or reviewed drafts of the paper, and approved the final draft.
- Ketut Suastika conceived and designed the experiments, authored or reviewed drafts of the paper, and approved the final draft.
- Safarina G. Malik analyzed the data, conceived and designed the experiments, authored or reviewed drafts of the paper, and approved the final draft.

### Human Ethics

The following information was supplied relating to ethical approvals (i.e., approving body and any reference numbers):

Eijkman Institute Research Ethics Commission and Faculty of Medicine Ethic Committee, Universitas Udayana granted Ethical approval to collect samples and carry out the study (No. 32 on 27 October 2008 and No. 80 on 24 December 2014; No. 690a/SKRT/X/2010 on 28 October 2010 and No. 1286/UN.14.2/Litbang/2014 on 18 September 2014).

### Field Study Permissions

The following information was supplied relating to field study approvals (i.e., approving body and any reference numbers):

We received permission from the following leaders of villages and regencies: I Made Madia Suryanatha S.S.T.P (Legian Village of the Badung Regency), I Wayan Supat (Penglipuran Village of the Bangli Regency), Ketut Gede Arjaya (Nusa Ceningan Village of the Klungkung Regency) and I Putu Sudarmaja (Pedawa Village of the uleleng Regency).

### Data Availability

Data is available at Mendeley: Priliani, Lidwina (2019), "FTO dataset", Mendeley Data, v1. DOI 10.17632/5zczrdkfgh.1.

### Supplemental Information

Supplemental information for this article can be found online at http://dx.doi.org/10.7717/peerj.8327#supplemental-information.

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
