# Peer review of "Obesity in the Balinese is associated with FTO rs9939609 and rs1421085 single nucleotide polymorphisms"

_PeerJ, doi:10.7717/peerj.8327_

## Round 0.1 · original submission · Major Revisions

Dear authors,

In light of the reviewers' comments, I think your manuscript has scientific merit to be published in PeerJ. However, there are some issues which you must apply in a revised version of the text.

Best regards,
Dr Palazón-Bru (academic editor for PeerJ)

Reviewer 1 ·

Basic reporting

Priliani and collaborators examined the associations of FTO rs9939609 and rs1421085 SNPs with obesity in the 612 unrelated Balinese subjects living in urban and rural areas. They found several significant associations with obesity anthropometric parameters (BMI, WC and WHtR) as also with high BMI (overweight/obesity), high WC and high WHtR.
The paper is interesting and well written. However, some results are hard to understand.

Experimental design

Methods described with sufficient detail & information to replicate.

Validity of the findings

The statistical analysis was done using linear regression, with adjustments for age, gender (male/female) and population (urban/rural). In Results section (L143-145) the authors claim that “Of the non-genetic parameters, age was shown to influence BMI, and being male increased BMI and WC, while living in an urban setting increased all obesity parameters” (data shown in Table 2). Also Tables 3, 5 and 6 shows that urban setting increase most obesity parameters in a statistically significant way. However, Table 4 shows that significant results were obtained for BMI in rural population, and the authors claim in Abstract that “The significant associations were particularly found in … rural population.” Thus, some kind of contradiction seems to exist between the obtained data. Or the covariates across all tables should be referred as gender and population (instead of male and urban, respectively)?

Additional comments

Minor points:

Change “females” by “women” and “males” by “men” along the manuscript.

Tables 2-6 report data obtained by linear and logistic regression. Hence, both models should be referred below the tables.

L52: “increased” rather than “increases”

L124: Consider “0.05/2” rather than “0.050/two SNPs”

L201: Consider rephrasing “that having the FTO rs9939609 and rs1421085 SNPs” to “that having the FTO rs9939609 and rs1421085 risk alleles”

L223-225: “Our results indicated the modulation by FTO rs9939609 and rs1421085 in maintaining higher BMI in older subjects, in contrast to the Balinese general characteristics where BMI tend to decrease by age.” Consider rephrasing this sentence since it seems confuse…

L228: “more” instead “mroe”

L229: Consider rephrasing “obsese students” by "students having obesity"

Reviewer 2 ·

Basic reporting

It was clear and well structured.

Experimental design

The design was good. I have some notes about details for information that is being provided in the general comments.

Validity of the findings

No comment

Additional comments

Review PeerJ
Overall, this is an interesting study, well written and advanced statistical analysis.

Introduction
It will be interesting to put the prevalence of obesity in Bali compare to national data based on RISKESDAS 2018.

Tourism might play a role in speeding up a modernity in a population. And this is not exclusive to Bali. It might be important that author highlight the impact of urban development on obesity (which Jakarta being the important case), however repeating role of tourism might at the introduction might lead to an assumption that tourism is to be blame for increasing rate of obesity in Bali. My suggestion is that Author can suggest economic development induces obesity without necessarily point finger that tourism industry is the sole factor.

Methods
Can author describe the inclusion and exclusion criteria for subjects? What is the maximum age for subjects to be involved.

Results
To me, it is unclear why in the title and context, author used obesity as an outcome, while in the analysis authors used “high BMI” which means BMI higher than 25, rather than the actual cut off for obesity.

Table 4.
When looking at the unadjusted estimates and odds ratio, the correlation between FTO and high BMI was some significant in rural rather than in rural. Is there any explanation. What is the differences between rural and urban adults in Bali, in respect to their diet/physical activity or other obesity-related factors?

The role of age.
Authors divide the group based on the mean age of the population. How old is the oldest subject? Can author describe the distribution of the number of subjects in each group?

Discussion
The author needs to be careful when mentioning "older subjects” as in this study it meant “subjects whose age higher than the mean age of study population”. The terminology of older adults is not that clear, however, WHO suggested anyone above 60 years old (https://www.who.int/healthinfo/survey/ageingdefnolder/en/). The author also might need to explain that BMI 30 in adults and older adults is different because the older we are, the higher fat percentage we have in our body (and less muscle). This issue can be avoided by measuring body fat.

The differential effect of FTO on obesity parameters between urban and rural is indeed the unique point of this study. It is unfortunate that Authors did not have records on lifestyle factors within those study populations. However, Authors might need to explain more about the difference in lifestyle between Balinese who live in rural and urban, based on previous findings by other studies. This could help the reader understand the context of gene-environment interaction that might be involved in this study.

It is important that the author also explains the mechanism of action of FTO on obesity. One aspect that was not really explained in this paper was the influence of FTO on appetite regulation. This was confirmed by appetite parameter or hunger/satiety hormones. Which might be the difference between gender/ or this influence might be modified by the environment.

Can the author suggest the implication of this study for developing personalized nutrition advise for the case of obesity?


Line 228 “more”

·

Basic reporting

The manuscript is clear and well-written. Appropriate references are cited.

Experimental design

Well-defined research question and clear methods and experimental approach.

Validity of the findings

Findings appear to be robust. Clear conclusions based on the findings.

Additional comments

The goal of this study was to examine the association between two SNPs in the FTO gene and markers of adiposity in a Balinese population. The study is well-designed and the manuscript is clear and very well-written. The authors used appropriate methods and analyses and the findings show significant associations between FTO genotype and various parameters of adiposity across different demographics. I only have the following minor comments for the authors to consider:
1. In Figure 1, it is not clear how the top 3 figures differ from the bottom 3 figures. This should be marked in the figure legend and perhaps use letters to denote each panel (eg Panel A-C on top and panels D-F on the bottom).
2. Tables 5-7 show the results of haplotype analyses, however, it is not clear what the reference/comparison group is here. A clearer description of the analyses conducted here would be helpful. The table should clearly describe what the haplotype refers to. For example, rs9939609/rs1421085 should be noted in the column heading so that readers can clearly see that AC refers to nucleotides of the respective SNPs. Are these true haplotypes (ie nucleotides on the same strand), or are they combined genotypes? What exactly does AC represent? Does it mean they are carriers of the A and C alleles of the respective SNPs, or are they homozygotes for those SNPs (ie AA/CC)? Is there a TT haplotype? If the authors choose to remove the haplotype analyses (since the two SNPs are highly linked), I don’t think that would diminish their findings.

---

## Round 0.2 · accepted · Accept

All the reviewers' concerns have been correctly addressed.

Reviewer 1 ·

Basic reporting

no comment

Experimental design

no comment

Validity of the findings

no comment

Additional comments

I have no further comments on the manuscript.